# Heart Rate Variability as a Surrogate Marker of Severe Chronic Coronary Syndrome in Patients with Obstructive Sleep Apnea

**DOI:** 10.3390/diagnostics13172838

**Published:** 2023-09-01

**Authors:** Christopher Seifen, Maria Zisiopoulou, Katharina Ludwig, Johannes Pordzik, Muthuraman Muthuraman, Haralampos Gouveris

**Affiliations:** 1Sleep Medicine Center & Department of Otolaryngology, Head and Neck Surgery, University Medical Center Mainz, 55131 Mainz, Germany; kim.seifen@unimedizin-mainz.de (C.S.); katharina.ludwig@unimedizin-mainz.de (K.L.); johannes.pordzik@unimedizin-mainz.de (J.P.); 2Department of Cardiology, University Hospital Frankfurt, Goethe University Frankfurt am Main, 60629 Frankfurt am Main, Germany; maria.zisiopoulou@kgu.de; 3Neural Engineering with Signal Analytics and Artificial Intelligence, Department of Neurology, University Medical Center Würzburg, 97080 Würzburg, Germany; muthuraman_m@ukw.de

**Keywords:** obstructive sleep apnea, OSA, heart rate variability, HRV, chronic coronary syndrome, CCS, surrogate marker

## Abstract

Background and Objectives: Obstructive sleep apnea (OSA) is a known risk factor for chronic coronary syndrome (CCS). CCS and OSA are separately associated with significant changes in heart rate variability (HRV). In this proof-of-concept study, we tested whether HRV values are significantly different between OSA patients with concomitant severe CCS, and OSA patients without known CCS. Material and Methods: The study comprised a retrospective assessment of the historical and raw polysomnography (PSG) data of 32 patients who presented to a tertiary university hospital with clinical complaints of OSA. A total of 16 patients (four females, mean age 62.94 ± 2.74 years, mean body mass index (BMI) 31.93 ± 1.65 kg/m^2^) with OSA (median apnea-hypopnea index (AHI) 39.1 (30.5–70.6)/h) and severe CCS were compared to 16 patients (four females, mean age 62.35 ± 2.06 years, mean BMI 32.19 ± 1.07 kg/m^2^) with OSA (median AHI 40 (30.6–44.5)/h) but without severe CCS. The short–long-term HRV (in msec) was calculated based on the data of a single-lead electrocardiogram (ECG) provided by one full-night PSG, using the standard deviation of the NN, normal-to-normal intervals (SDNN) and the heart rate variability triangular index (HRVI) methods, and compared between the two groups. Results: A significant reduction (*p* < 0.05) in both SDNN and HRVI was found in the OSA group with CCS compared to the OSA group without CCS. Conclusions: Severe CCS has a significant impact on short–long-term HRV in OSA patients. Further studies in OSA patients with less-severe CCS may shed more light onto the involved mechanistic processes. If confirmed in future larger studies, this physiologic metric has the potential to provide a robust surrogate marker of severe CCS in OSA patients.

## 1. Introduction

Obstructive sleep apnea (OSA) is the most common type of sleep-disordered breathing. This condition is characterized by recurrent episodes of partial or complete airway obstruction during sleep, leading to repetitive apneas or hypopneas, even though respiratory effort is still present. The clinical signs and symptoms include sleep interruption, snoring, and daytime sleepiness, which can lead to a significant impairment in quality of life. Moreover, OSA is associated with an increased risk of cardiovascular morbidity and mortality [1], particularly an increased risk of coronary artery disease (CAD) and an increased risk of heart failure [2,3]. The term “CAD” covers a wide spectrum, from asymptomatic individuals to those suffering from acute coronary syndrome. CAD is considered a dynamic process with prolonged stable phases, which is why the term “stable CAD” was used for a long time. Nevertheless, stable CAD may become unstable, leading to a sudden acute coronary event, such as acute coronary syndrome. Therefore, the term “stable CAD” has, most recently, been replaced by “chronic coronary syndrome” (CCS) [4].

The mechanisms for cardiovascular disease development in OSA are complex, and include sympathetic nervous system overactivity, endothelial dysfunction, inflammation, and oxidative stress [5]. In moderate-to-severe OSA, an overactivity of the sympathetic nervous system is considered one of the main factors in the development of CCS [6]. CCS is a major public health problem, accounting for the majority of deaths in the United States [7]. According to this trend, half of all healthy 40-year-old men will develop CCS in the future, and one in three healthy 40-year-old women [8].

The analysis of heart rate variability (HRV) from electrocardiography (ECG) signals provides an opportunity to assess shifts in the autonomic nervous system in a noninvasive and cost-effective manner. HRV reflects the adaptive capacity of the cardiovascular system in response to various stressors; therefore, HRV is considered to be a very clinically relevant parameter, from a wide variety of perspectives. HRV is characterized as the variability in the beat-to-beat intervals of the heart, and is typically measured using RR intervals extracted from an ECG signal. Each RR interval represents the time elapsed between successive R peaks of the QRS complex on the ECG signal. Fluctuations in RR intervals are mediated via sympathetic and vagal efferent activity, and can be affected by physiological, pathological, physical, and psychological activity [3]. HRV values have been shown to be low in patients with CCS, and low HRV has been shown to be an independent predictor of cardiovascular mortality and sudden cardiac death [9,10,11,12,13,14,15]. Additionally, a recent meta-analysis found that reduced HRV was associated with a 46% greater risk of cardiovascular events [16]. Thus, there is increasing evidence that HRV may indeed provide a surrogate marker of underlying cardiovascular risk.

In OSA patients, the use of HRV analysis provides information on cardiac autonomic control. HRV analyses in OSA patients show a shift in autonomic modulation toward the predominance of sympathetic activity and attenuated parasympathetic activity. While the severity of OSA is usually determined via the apnea–hypopnea index (AHI), additional physiological metrics in combination with the AHI, such as the HRV, may help to further refine the characterization and phenotyping of OSA disease states and OSA disease severity. For instance, ultra-short-term HRV tends to increase with the increasing duration of the respiratory events (apneas and hypopneas) during sleep [17]. In addition, the short-term HRV response differs as a function of the desaturation severity and the respiratory event rate in patients with suspected OSA [18]. Moreover, other authors found lower high-frequency power in normalized units, and higher low-frequency power in normalized units, in severe OSA patients, compared to individuals without OSA [19]. The same authors have found that the standard deviation of normal RR intervals (SDNN), and the root mean square of successive RR interval differences, were independently associated with OSA severity [19]. Various existing applications of HRV in different aspects of OSA have already been carried out, to examine the impaired neuro-cardiac modulation in OSA [20].

Based on these findings, the present study was intended to be a proof-of-concept study to investigate whether HRV-related metrics were significantly different between OSA patients with severe CCS, and OSA patients without CCS. We hypothesized that HRV measurements in patients with severe CCS would differ significantly from those without CCS, and that the short–long-term HRV metric might provide a surrogate marker for severe CCS, specifically in patients with OSA.

## 2. Materials and Methods

### 2.1. Study Protocol

The clinical database of our sleep laboratory of a tertiary university medical center was searched retrospectively, between January 2019 and April 2021, for patients with clinical complaints of OSA undergoing first-time polysomnography (PSG) according to the American Academy of Sleep Medicine’s (AASM, Inc., Darien, IL, USA) standard guidelines [21]. All patients enrolled in the study received first-time PSG to screen for OSA, based on a history of snoring and/or witnessed apneas and/or daytime sleepiness; in other words, the OSA screening was due to sleep disorder symptoms, and not due to a routine health maintenance examination, or high-risk screening. Each PSG was conducted overnight by a licensed technician, and interpreted by a board-certified sleep specialist.

The baseline evaluation of all patients included the assessment of demographic characteristics, e.g., age, sex, and body mass index (BMI). An extended search included aspects that may have a complementary confounding influence on sleep and, hence, on the PSG-based HRV metric, e.g., daily alcohol consumption, or primary nighttime work (e.g., patients working a night shift). In addition, the medical files of all the patients were reviewed, to search for any cardiac history (e.g., a history of chronic coronary syndrome—CCS, a history of atrial fibrillation or other cardiac arrhythmia, a history of myocardial infarction, a history of percutaneous coronary intervention with stenting (PCI-S) or a history of coronary artery bypass grafting (CABG), and a history of non-cardiac diseases (e.g., arterial hypertension, diabetes mellitus, chronic obstructive pulmonary disease—COPD, or bronchial asthma). All patients included in the study were aged 18 years or older. Patients with the presence of a pacemaker or implantable cardiac defibrillator (ICD), as well as patients with >25% central sleep apnea events, were excluded from the study.

Two groups were formed: (1) all male and female patients with a CCS history were assigned to the group “CCS”, namely “chronic coronary syndrome”; and (2) all male and female patients without a cardiac history were assigned to the group “CON”, namely “control”. After the evaluation of a total of 483 consecutive patients, 16 patients were found eligible to form the CCS group. From the remaining 467 (=483 − 16) patients, we found a respective group of OSA patients who were the best possible match, according to the exclusion and inclusion criteria mentioned above. All patients assigned to the group “CCS” were manually selected from the clinical database of all the appropriate first-time PSGs in our sleep laboratory. Through this process, we ensured that all the patients had a securely documented history of CCS. Here, we relied on previous in-house written (diagnostic) documents or submitted external physician reports. To improve the quality of our patient selection, we extended the period of data collection, as mentioned above. As a result, the selected group is a very specific group, namely OSA patients with chronic coronary syndrome. To form the CON group, we considered only the most eligible patients with respect to the main clinically relevant endpoints; namely, age, sex, BMI, AHI, and the above-mentioned non-cardiac diseases, as well as aspects disturbing sleep (alcohol consumption, nighttime work); from the clinical database of our sleep laboratory, to avoid significant differences between the two study groups. The comparison between the study groups/participants is shown in Table 1 and Table 2.

### 2.2. Calculation of Heart Rate Variability

Short–long-term HRV (in msec) was calculated using the raw data of a single-lead electrocardiogram (ECG) recording, as provided for one full PSG night, according to the American Academy of Sleep Medicine’s (AASM, Inc., Darien, IL, USA) standard guidelines [21]. The calculation of HRV was performed as described in [22]. All the ECG recordings were saved at a sampling frequency of 1 kHz (signal bandwidth 0.04–387 Hz) and a resolution of 1 μV/bit. More specifically, the SDNN is the standard deviation of the NN, normal-to-normal (RR) intervals. The estimation of the heart rate variability triangular index is divided into three steps, as described via a representative example in Figure 1. The first step is to define the RR intervals, as shown in the first plot for three heart beats. Once the RR intervals are estimated, during the second step, the corresponding tachogram of the RR intervals is computed, as shown in the second plot for this representative example. In the third step, the distribution of the duration of the normal RR intervals is computed, and the X representative bin is defined; additionally, the number of normal RR intervals, respectively, in that bin (Y) is found, as shown in the third plot in Figure 1. At the end, the HRV index was defined as the total number of NN intervals divided by the number of NN intervals in the representative bin and, later, the findings were compared between the two OSA patient groups [22].

### 2.3. Support Vector Machine Analyses

To investigate the significance of these short–long-term HRV metrics, we used a support vector machine (SVM) algorithm to classify the two groups based on the two short–long-term HRV metrics. In brief, SVM is a powerful tool for the non-linear classification between two datasets, which looks for an optimal separating threshold between the two datasets, by maximizing the margin between the classes’ closest points [23]. Here, we used the polynomial function kernel for this projection, due to its good performance, as previously discussed, and used the grid search (min = 1; max = 10) to find the few optimal input parameters and gamma (0.25). The selection was checked via 10-fold cross-validation, using 75% of the data for training, and 25% for testing. Hence, the classification accuracy was estimated for the training, testing, and overall set of data after 10-fold cross-validation.

### 2.4. Ethical Statement

All patients had provided informed consent for the use of their data for research purposes. The data were evaluated in a pseudonymized fashion. Due to this fact, and the retrospective nature of the study, the local institutional review board (IRB) was consulted. A separate approval was waived by the local IRB, because all retrospective study procedures in this study were in accordance with local data protection and research practices regarding humans. All procedures were in full accordance with the Declaration of Helsinki.

### 2.5. Statement about Statistics and Calculation

All the epidemiological/demographic, clinical, and PSG-based metric data (e.g., AHI) were statistically analyzed using JMP 14 (SAS Institute, Cary, NC, USA) and GraphPad Prism version 5.01 (GraphPad Software, Boston, MA, USA). Categorical variables were described as number and percentage (%), and continuous variables were described as mean ± standard deviation (SD) for normally distributed values, or median and interquartile range (IQR) for non-normally distributed values. All statistical tests were performed after the evaluation of the normality of distribution via the Shapiro–Wilk test and Kolmogorov–Smirnov test. Comparisons between the groups were analyzed using the *t* test for normally distributed values, and the Mann–Whitney test for non-normally distributed values. The results were considered significant when the *p*-value was <0.05.

A data-driven regression model was implemented, without the explicit stating of a functional form, indicating a nonparametric technique. The procedure was quite similar to the one used by Malatantis-Ewert et al. [24]. The algorithm looks for an optimal separating threshold between the two data sets by maximizing the margin between the classes’ closest points. The points lying on the boundaries are called support vectors, and the middle of the margin is the optimal separating threshold. In most cases, the linear separator is not ideal; therefore, a projection into a higher-dimensional space is performed, where the data points effectively become linearly interrelated. Here, we have used the RBF kernel for this projection, due to its good performance, as discussed by Cortes and Vapnik (1995), and based on the previous application of support vector machines in earlier studies [23,25,26]. We then used the grid search (min = 1; max = 10) to find the few optimal input regularization parameters, namely C (the type of classification algorithm), which is the capacity constant. The parameter C should be carefully chosen, because the larger the C, the more the error is penalized (i.e., it leads to over-fitting), so we tested values in the range of 1–1000, and chose a gamma of 0.25 for the RBF kernel function (which represents the data for cross-validation). The selection was checked via 10-fold cross-validation, through 75% of the data set being taken for training, and 10% for testing. A soft-margin classifier of the calculated independent variables was used for every parameter, and spurious correlations (correlations which could be due to artifacts) were weighted by a penalty constant, P. To optimize the correlation coefficient, this was calculated for every regressor. To demonstrate that no over-fitting is attested in our data for the SVM regression algorithm, we performed cross-validation. The results from the SVM are reported here with 10-fold cross-validation.

## 3. Results

### 3.1. Demographic Characteristics and Severity of Obstructive Sleep Apnea

A total of 12 male patients (75%) and 4 female patients (25%) were assigned to the group CCS. The age distribution in this group was 43–78 (62.94 ± 2.74) years, and the BMI was 31.93 ± 1.65 kg/m^2^. Two patients (12.5%) consumed alcohol daily, and two patients (12.5%) performed shift work during the night. The AHI of this group was 39.1 (30.5–70.6)/h.

Accordingly, 12 male patients (75%) and 4 female patients (25%) were assigned to the group CON. The age distribution in this group was 41–76 (62.35 ± 2.06) years, and the BMI was 32.19 ± 1.07 kg/m^2^. Two patients (12.5%) consumed alcohol daily, and two patients (12.5%) performed shift work during the night. The AHI of this group was 40 (30.6–44.5)/h.

The comparison of age, BMI, and AHI between the two groups revealed no significant differences (*p* = 0.8650, *p* = 0.8966, *p* = 0.6923, respectively). Similarly, the number of patients with daily alcohol consumption or nighttime employment was identical. All demographic characteristics and the severity of OSA are shown in Table 1.

### 3.2. Medical History of the Study Groups

All patients in the CCS group had chronic coronary syndrome. In addition, patient medical files revealed that 4 (25%), 2 (13%), 9 (56%), 9 (56%), and 2 (13%) patients, respectively, had a history of atrial fibrillation (AF) or other (non-AF) cardiac arrhythmias, a history of myocardial infarction, a history of PCI-S, or a history of CABG. Further, in this group, arterial hypertension, diabetes mellitus, COPD, and bronchial asthma were present in 9 (56%), 3 (19%), 1 (6%), and 2 (13%) patients, respectively.

In the CON group, there was no recorded history of cardiac disease. In this group, arterial hypertension, diabetes mellitus, and bronchial asthma were present in 11 (69%), 4 (25%), and 2 (13%) patients, respectively. Detailed information about the diseases that could be found within the patients’ medical files are listed in Table 2.

### 3.3. Comparison of Short–Long-Term Heart Rate Variability

The SDNN and the HRV index showed a clear significant decrease (*p* < 0.05) in the OSA patients with CCS (“Patients”) compared to the OSA patients without CCS (“Controls”). In addition, the SVM results validated the above significant results through the use of these two parameters. The two groups can be classified with a median overall accuracy of 70%. Even though Table 1 reflects no differences in the demographics of these two groups of patients (OSA with and without CCS; in other words, “CCS” vs. “CON”), the two HRV metrics appear to be very robust in distinguishing the two groups (see Figure 2).

## 4. Discussion

### 4.1. Key Findings

In this proof-of-concept study, a significant decrease in the standard deviation of NN intervals (SDNN) and the heart rate variability triangular index (HRV index) was found in the OSA patients with severe CCS, compared to the OSA patients without CCS. To our knowledge, this is the first proof-of-concept study providing such evidence.

### 4.2. Comparison with Other Studies of Heart Rate Variability in Chronic Coronary Syndrome Patients

Previously, data from a large prospective clinical study had shown that short-term (1-h) HRV testing could be used for an enhanced risk assessment in low-to-intermediate-risk individuals without known CCS. In that study, participants had undergone 1 h Holter testing, with immediate HRV analysis, using the HeartTrends DyDx algorithm [27]. Low HRV, assessed using the HeartTrends DyDx algorithm, was shown to be independently associated with a 2-fold increased risk for the presence of myocardial ischemia in individuals without known CCS [27]. Nonetheless, no information on the presence of OSA in their cohort was presented by these authors.

In another significant development in this field, Shi et al. [28] proposed a novel entropy metric called Rényi distribution entropy (RdisEn) in the analysis of short-term heart rate variability signals and the detection of CCS. In addition, compared to the sample entropy or approximation entropy outputs, the RdisEn output was less affected by the parameter choice, and it remained stable even in short-term HRV. By developing a combined CCS detection scheme with RdisEn and wavelet packet decomposition, and applying k-nearest neighbor and SVM classifiers to separate CCS patients from normal subjects, the authors’ scheme automatically detected CCS with 97.5% accuracy, 100% sensitivity, and 95% specificity, providing an impressive detection performance. Nonetheless, no OSA patients, or information on OSA, were included in this study.

In patients with CCS who have undergone CABG surgery, the baroreceptor sensitivity, obstructive and central apnea indices, desaturation index, and lowest O2 saturation were not significantly different between the preoperative values, day 6, and day 30 after surgery [29]. Based on these findings, one may argue that the underlying pathophysiologic processes controlling HRV in OSA patients may be quite robust, and may not be influenced by interventions (e.g., surgery) for concomitant OSA and CCS. If this hypothesis is true, as suggested by the results from Al Hashmi et al. [29], then the results presented in our present study may not be confounded by respective performed interventions to treat CCS. This issue should, of course, be investigated in future studies.

### 4.3. Strengths and Limitations of This Study

This is one of the very few controlled studies investigating HRV in concomitant CCS and OSA using very-well-matched patient cohorts. Despite an extensive literature search, we failed to identify any similar study. The two patient groups were very well described regarding clinical and PSG-related characteristics, and they were very successfully matched regarding major epidemiologic confounders, namely sex, age, BMI, and even their degree of OSA severity, based on the established AHI metric. In addition, comparable behavior was found in terms of daily alcohol consumption, and a comparable number of patients performing mainly shift work at night. Accordingly, both groups had comparable non-cardiac conditions, namely arterial hypertension, diabetes mellitus, COPD, or bronchial asthma. This fact ensures the minimization of potentially confounding epidemiological metrics. Nevertheless, we must acknowledge that our study population is rather small. Another limitation is that the selection of patients from our clinical database was done manually, against the background of the most suitable patients for this study. Therefore, a potential selection bias cannot be excluded with certainty, and the selection may have also influenced the results presented. To minimize this major limitation, future studies should consider larger patient cohorts that are randomly selected, and automatically matched for epidemiologic confounders.

It should be, once again, stressed that the present study was intended as a preliminary proof-of-principle study. Therefore, the study population is rather small. Nonetheless, the results appear to be quite robust, even with this rather small sample size. A sensitive majority of widely used medications may affect sleep individually. The theory cannot be excluded that medication used on a regular basis by the patients, especially in the OSA patient group with cardiac history (e.g., beta-blockers), may have confounded the HRV measurements and, hence, the results. Further studies should specifically consider this aspect. Possible additional confounding factors on the HRV measurements may also have been the various interventions (e.g., PCI, CABG) for the treatment of severe CCS that had been performed before the PSG measurements in our studied severe CCS patient cohort. It should be noted that the OSA patients in our cohort without known or symptomatic CCS had not previously undergone coronary angiography or other composite coronary artery diagnostics; therefore, we cannot exclude with certainty the concomitant presence of subclinical (although, in any case, non-symptomatic) CCS in this cohort of patients in our study.

### 4.4. Perspective and Clinical Significance

These results from a single center can only be considered preliminary, and should be further validated. However, one potential application of the results presented is to augment the general screening for changes in HRV during diagnostic PSG in the sleep laboratory. Even if the exact source of HRV changes after PSG cannot be identified, extended screening for the presence of CCS might be useful if the risk profile is appropriate. This approach may help to reduce the risk of cardiovascular morbidity and mortality due to CCS. Thus, HRV analysis can aid in enhancing preventive care strategies, and improving public health.

Heart rate variability analysis using electrocardiography (ECG) signals is a physiological metric that has been established to assess changes in autonomic nervous system activity at large. It is non-invasive, and rather cost-effective [3]. Variability in RR intervals is modulated and effected simultaneously by both sympathetic and vagal efferent activity, and can be affected by apneas and hypopneas during sleep. When sympathetic activity is upregulated, and parasympathetic activity is downregulated, as is the case in OSA, the heart rhythm tends to converge toward the intrinsic heart rate physiological attractor (namely, around 100 heart beats/min), culminating in a reduced HRV [30]. In addition, an increase in the value of the SDNN as a surrogate marker of global autonomic function has been previously reported to be associated with better cardiovascular outcomes [31]. Especially regarding CCS, a low HRV has been independently associated with a significant 2-fold-increased likelihood for myocardial ischemia in patients without OSA [27].

On the basis of our preliminary results, we suggest that further studies should be performed with much larger, and also well-matched cohorts of patients with OSA, but without CCS, and patients with concomitant OSA and CCS. Further, other studies of our proposed HRV metrics in OSA patients with less severe CCS (e.g., no history of PCI, or no history of myocardial infarction) should be performed to shed more light on the pathophysiological mechanisms involved. If these physiological HRV metrics are confirmed in future studies with substantially more participants, then they have the potential to provide robust surrogate markers for severe CCS in OSA patients.

In addition, it should be also further tested whether the observed differences in HRV metrics in OSA patients are CCS-specific (and not heart-disease-associated, in general). To this end, further investigations in OSA patients with other cardiovascular concomitant diseases should be performed and HRV metrics should be compared among the various comorbid OSA–cardiac-disease cohorts.

Another major issue involves the possible differential effects of gender on HRV metrics in OSA patients, given that there is accumulating evidence that gender has a significant impact on the pathophysiology of OSA [32,33].

We suggest that such HRV metrics can also be used in conjunction with more traditional/established cardiovascular risk factors to identify individuals without known CCS who have an increased likelihood of the presence of myocardial ischemia [27]. In addition, the influence of the various pharmacological and interventional or surgical procedures on such HRV metrics in OSA patients with CCS should be the topic of future investigations.

## 5. Conclusions

We present preliminary evidence from a controlled study, with well-matched cohorts, that HRV metrics can differentiate between OSA patients with comorbid severe CCS, and OSA patients without concomitant CCS.

## Figures and Tables

**Figure 1 diagnostics-13-02838-f001:**
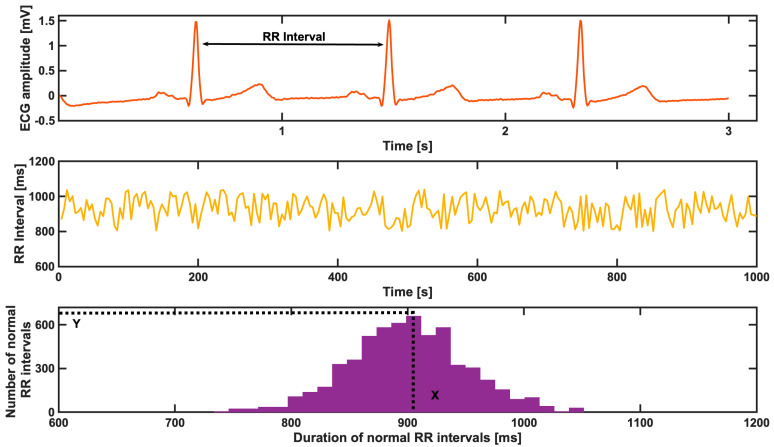
The first plot shows the raw electrocardiogram trace for 3 s, as a representative example. The second plot shows the tachogram of the RR intervals for an example ECG trace part, encompassing a time window of 1000 s. The third plot shows the distribution of the duration of the RR intervals, and indicates the X defined as the representative bin. Y indicates the total number of normal RR intervals.

**Figure 2 diagnostics-13-02838-f002:**
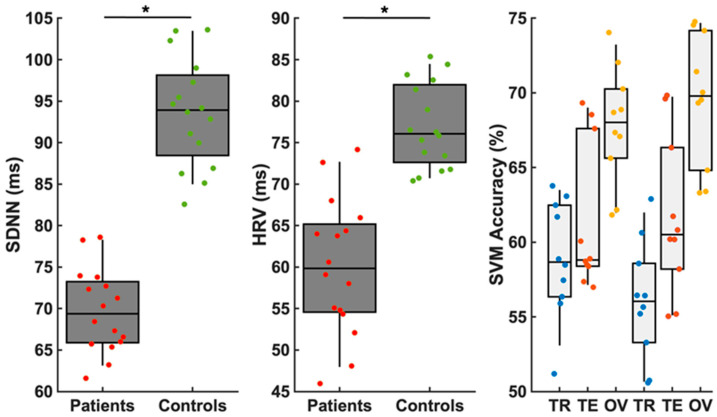
Box plots presenting the distribution of short–long-term HRV (in milliseconds; ms) using the standard deviation of normal RR intervals (SDNN) and heart rate variability (HRV) index methodology in OSA patients with (red points), and without (green points) chronic coronary syndrome (CCS). The quite-significant decrease (*p* < 0.05) in the HRV metrics in the OSA patients with CCS, compared to the OSA patients without CCS, can be appreciated. The support vector machine (SVM) accuracy is shown for the training (TR, blue dots), testing (TE, red dots) and overall (OV, yellow dots) data for each of the cross-validations separately. The asterisk (*) marks a statistical significance with *p* < 0.05.

**Table 1 diagnostics-13-02838-t001:** Demographic characteristics and the severity of obstructive sleep apnea in the study groups.

	CCS	CON	Between-Group Comparison (*p*-Value)
Number of patients	16	16	
Number of males (%)	12 (75)	12 (75)	
Patient with daily alcohol consumption (%)	2 (12.5)	2 (12.5)	
Patient working shift jobs overnight (%)	2 (12.5)	2 (12.5)	
Age, in years ± SD	62.94 ± 2.74	62.35 ± 2.06	0.8650
BMI in kg/m^2^ ± SD	31.93 ± 1.65	32.19 ± 1.07	0.8966
AHI (*n*/h) (IQR)	39.1 (30.5–70.6)	40 (30.6–44.5)	0.5095

Abbreviations (in alphabetical order): AHI, apnea–hypopnea index; BMI, body mass index; CCS, patients with chronic coronary syndrome; CON, patients without any cardiac history; IQR, interquartile range; SD, standard deviation. Categorical variables are described as number and percentage (%), and continuous variables are described as mean ± SD for normally distributed values, or median and IQR for non-normally distributed values. The results were considered significant when the *p*-value was <0.05.

**Table 2 diagnostics-13-02838-t002:** Major comorbidities in the two study groups.

		CCS	CON
Cardiac history	CCS (*n* (%))	16 (100)	-
	Atrial fibrillation (AF) (*n* (%))	4 (25)	-
	Other (non-AF) cardiac arrhythmias (*n* (%))	2 (13)	-
	Myocardial infarction (*n* (%))	9 (56)	-
	PCI-S (*n* (%))	9 (56)	-
	CABG (*n* (%))	2 (13)	-
Non-cardiac history	Arterial hypertension (*n* (%))	9 (56)	11 (69)
	Diabetes mellitus (*n* (%))	3 (19)	4 (25)
	COPD (*n* (%))	1 (6)	-
	Bronchial asthma (*n* (%))	2 (13)	2 (13)

Abbreviations (in alphabetical order): CABG, a history of coronary artery bypass grafting; CCS, patients with chronic coronary syndrome; CON, patients without a history of CCS; COPD, chronic obstructive pulmonary disease; PCI-S, a history of percutaneous coronary intervention with stenting. Categorical variables are described as number and percentage (%).

## Data Availability

The data presented and analyzed in this study are available on reasonable request from the corresponding author.

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
