# Peer review of "Heart Rate Variability as a Surrogate Marker of Severe Chronic Coronary Syndrome in Patients with Obstructive Sleep Apnea"

_diagnostics, 2023, doi:10.3390/diagnostics13172838_

Round 1
Reviewer 1 Report
This paper investigate HRV parameters in OSA patients with and without chronic coronanry syndrome. This is an interesting issue in clinical practice. Though there are some limitation in this study, the authors have clearly pointed out and give directions for future study. However, I think there still some part need to be improved but better understanding. Firstly, in the section 2.2.1 Calculation of heart rate variability, it is difficult for readers to understand how HRV index is calculated and what HRV index represented. A figure illustrating this calculating and using your own sentence (rather than copy from the reference) might be helpful. Secondly, the discussion part is relative insufficient. Readers might wondering why SDNN and HRV index decrease in CCS group. A description adrresing the rationale underlying automonic nervous system, coronary artery disease and HRV might be helpful.
Author Response
This paper investigate HRV parameters in OSA patients with and without chronic coronanry syndrome. This is an interesting issue in clinical practice. Though there are some limitation in this study, the authors have clearly pointed out and give directions for future study. However, I think there still some part need to be improved but better understanding.
Firstly, in the section 2.2.1 Calculation of heart rate variability, it is difficult for readers to understand how HRV index is calculated and what HRV index represented. A figure illustrating this calculating and using your own sentence (rather than copy from the reference) might be helpful.
Response: thank you for this comment. We have now added both a more detailed description of the HRV index calculation and a respective figure (Figure 1 in the revised manuscript) illustrating this calculation, as suggested by our reviewer.
Secondly, the discussion part is relative insufficient. Readers might wondering why SDNN and HRV index decrease in CCS group. A description adrresing the rationale underlying automonic nervous system, coronary artery disease and HRV might be helpful.
Response. Thank you very much for this remark, which will definitely improve comprehension by the non-expert readers. We added the following part to the “Discussion” part under the heading “Perspective and Clinical Significance”:
“Heart rate variability analysis of electrocardiography (ECG) signals is a physiological metric that has been established to assess changes in autonomic nervous system activity at large. It is non-invasive and rather cost-effective (Ucak et al., 2021). Variability in RR intervals is modulated and effected simultaneously by both sympathetic and vagal efferent activity and can be affected by apneas and hypopneas during sleep. When sympathetic activity is upregulated and parasympathetic activity is downregulated, as is the case in OSA, the heart rhythm tends to converge towards the intrinsic heart rate physiological attractor (namely, around 100 heart beats/min) culminating to a reduced HRV (Opthof, 2000). In addition, an increase in the value of the SDNN, being a surrogate marker of global autonomic function, has been previously reported to be associated with better cardiovascular outcomes (Hillebrand et al., 2013). Specifically regarding CCS, low HRV has been independently associated with a significant 2-fold increased likelihood for myocardial ischemia in patients without OSA (Goldenberg et al.2019)”
As a result, we have now added two additional references in the reference list, namely:
Hillebrand, S., Gast, K. B., de Mutsert, R., Swenne, C. A., Jukema, J. W., Middeldorp, S., Rosendaal, F. R., & Dekkers, O. M. (2013). Heart rate variability and first cardiovascular event in populations without known cardiovascular disease: Meta-analysis and dose–response meta-regression. Europace, 15(5), 742–749. https://doi.org/10.1093/europ ace/eus341
Opthof, T. (2000). The normal range and determinants of the intrinsic heart rate in man. Cardiovascular Research, 45(1), 177–184. https:// doi.org/10.1016/S0008-6363(99)00322-3
The papers by Ucak et al. 2021 and Goldenberg et al. 2019 were already included in the former reference list.
Thank you for carefully reading our manuscript.
Reviewer 2 Report
I would like to thank you the authors for the really interesting topic, which addresses the interplay between OSA and cardiovascular risk. Hereby my comments:
1 - I do not understand the selection process of the included patients completely. Did you choose the control patients manually? Would a matching-based on the mentioned variables not have been better, as in a case-control study? Was the inclusion done on a sequential basis? How can you account for potential selection bias?
2 - Were data on alcohol consumption and sleep-influencing medications available? And what about data on shift jobs overnight? Are there any data on CPAP adherence? This are all factors that potentially influence the measured outcome. If not included in the data, this needs to be specified in the limitations section.
3 - A comment on the general applicability of the findings: usually, patients with OSA are patients at cardiovascular risk due to a composite clinical situation (e.g. obesity, metabolic syndrome) and are therefore already screened for cardiovascular and/or metabolic diseases. Moreover, the HRV is highly influenced by many different cardiac conditions (and OSA as well) - so how can the proposed predictor differentiate between the source of the elevated HRV? And if this is not possible, is it cost-effective to recommend further cardiovascular screening in patients resulting at risk after the proposed HRV analysis?
4 - Is the HRV difference, although statistically significant, clinically relevant?
Author Response
I would like to thank you the authors for the really interesting topic, which addresses the interplay between OSA and cardiovascular risk. Hereby my comments:
1 - I do not understand the selection process of the included patients completely. Did you choose the control patients manually? Would a matching-based on the mentioned variables not have been better, as in a case-control study? Was the inclusion done on a sequential basis? How can you account for potential selection bias?
Response: Thank you for your suggestion to be more precise regarding this issue. Suitable patients were selected and included in this study after manual/visual review of the database of our sleep laboratory by the participating investigators. In this way, proper patients could be filtered out for the CCS group, e.g., those with a documented medical history of myocardial infarction or stent implantation, etc. After 16 matched patients had been selected for group CCS, manual selection was also used to identify 16 matched patients for the control group. At this point, in addition to the absence of cardiac history, it was necessary to form such a patient group that was almost identical in terms of age, BMI, and gender distribution. Undoubtedly, this is one major limitation of this study that -probably- has caused selection bias, but on the other hand enabled control of major clinical confounders. We mentioned a potential selection bias in the limitations part of the discussion of our manuscript (see “4. Discussion, Strengths and limitations of this study”).
2 - Were data on alcohol consumption and sleep-influencing medications available? And what about data on shift jobs overnight? Are there any data on CPAP adherence? This are all factors that potentially influence the measured outcome. If not included in the data, this needs to be specified in the limitations section.
Response: Thank you for this significant comment. You asked for data on alcohol consumption, sleep-influencing medication, and shift jobs overnight. Based on your suggestion, we rescanned extensively the charts of the included patients and, interestingly, found that an identical number of patients in both groups consumed alcohol on a daily basis and performed shift work during the night (see “2.1 Study protocol”, “3.1 Demographic characteristics and severity of obstructive sleep apnea” and Table 1). Indeed, various medications can also influence sleep. Although we were able to find information on medication schedules for a large proportion of patients, it was not possible to say with certainty whether the medication indicated was actually taken daily and permanently or was fully documented. We therefore decided not to make any statement on the permanent medication of the included patients and to include this topic in the limitations of the study (see “4. Discussion, Strengths and limitations of this study”). Regarding your query about CPAP adherence, it must be stated here that only diagnostic PSGs were included in the current analysis and that data based on subsequent OSA therapies of any kind have been not considered.
3 - A comment on the general applicability of the findings: usually, patients with OSA are patients at cardiovascular risk due to a composite clinical situation (e.g. obesity, metabolic syndrome) and are therefore already screened for cardiovascular and/or metabolic diseases. Moreover, the HRV is highly influenced by many different cardiac conditions (and OSA as well) - so how can the proposed predictor differentiate between the source of the elevated HRV? And if this is not possible, is it cost-effective to recommend further cardiovascular screening in patients resulting at risk after the proposed HRV analysis?
Response: We absolutely agree with your statement about OSA patients - a patient group with particularly increased cardiovascular risk. However, it is precisely because of the comorbidity of OSA and cardiovascular diseases that we consider screening for the presence of such diseases to be very important. Exactly this fact has also been the reason to match the two studied groups specifically according to their AHI and BMI. We are convinced that the PSG raw data can provide much more information about the patients’ physiological status than has been known to date. The PSG-based analysis of HRV could be one such targeted PSG-based metric. One perspective would be for the PSG to be used to scan for other health aspects in addition to sleep medicine data, e.g., in the form of screening for the risk of chronic coronary syndrome. We are not claiming that the exact source of an HRV change can be identified using PSG, but we advocate with our present report that an extended screening for the presence of chronic coronary syndrome could be meaningful if the risk profile is appropriate. We can not answer the question whether this screening strategy is cost effective from a public health policy perspective based on our present findings. Thank you for providing the impetus to discuss this aspect (see “4. Discussion, Perspective and Clinical Significance”).
4 - Is the HRV difference, although statistically significant, clinically relevant?
Response: Because abnormal HRV patterns are associated with disease, including chronic coronary syndrome, HRV analysis can aid in disease diagnosis or risk stratification. Not only can the HRV difference indicate cardiovascular risk, but it also provides valuable insight into the functioning of the autonomic nervous system, which directly affects vital body functions such as heart rate, blood pressure, and respiration. Against this background, the HRV difference is considered clinically relevant. We added a statement about its clinical relevance (see “1. Introduction”).
Again, thank you for carefully reading our manuscript.
Round 2
Reviewer 2 Report
I would like to thank the authors for providing an extensive comment section, that they implented in the manuscript. I feel it has improved in quality as a proof-of-concept study. I do not have any further comments.